# That Sounds Right: Auditory Self-Supervision for Dynamic Robot Manipulation

**Abitha Thankaraj**
New York University
abitha@nyu.edu

**Lerrel Pinto**
New York University
lerrel@cs.nyu.edu

**Abstract:** Learning to produce contact-rich, dynamic behaviors from raw sensory data has been a longstanding challenge in robotics. Prominent approaches primarily focus on using visual and tactile sensing. However, pure vision often fails to capture high-frequency interaction, while current tactile sensors can be too delicate for large-scale data collection. In this work, we propose a data-centric approach to dynamic manipulation that uses an often ignored source of information – sound. We first collect a dataset of 25k interaction-sound pairs across five dynamic tasks using contact microphones. Then, given this data, we leverage self-supervised learning to accelerate behavior prediction from sound. Our experiments indicate that this self-supervised 'pretraining' is crucial to achieving high performance, with a 34.5% lower MSE than plain supervised learning and a 54.3% lower MSE over visual training. Importantly, we find that when asked to generate desired sound profiles, online rollouts of our models on a UR10 robot can produce dynamic behavior that achieves an average of 11.5% improvement over supervised learning on audio similarity metrics. Videos and audio data are best seen on our project website: https://aurl-anon.github.io/.

**Keywords:** Dynamic manipulation, Self supervised learning, Audio

## 1 Introduction

Imagine learning to strike a tennis ball. How can you tell whether your shot is getting better? Perhaps the most distinctive feeling is the crisp, springy boom – that just sounds right. It is not just playing tennis, for which audition provides a rich and direct signal of success; for example consider everyday tasks like unlocking a door, cracking an egg, or swatting a fly. Recent works in neuroscience [1, 2, 3] have found that sound enhances motor learning of complex motor skills such as rowing, where being able to listen to the sound of the rowing machine during learning improves a person's eventual rowing ability.

In the context of robotics, the learning of motor skills has often been centered around using visual observations as input [4, 5, 6]. While this has enabled impressive success in a variety of quasi-static robotics problems, visual observations are often insufficient in dynamic, contact-rich manipulation problems. One reason for this is that while visual data contains high amounts of spatial information, it misses out on high-frequency temporal information that is crucial for dynamic tasks. Further, dynamic contact can often be challenging to perceive using only vision. To address this challenge, we will need to embrace other forms of sensory supervision that can provide this information.

Using audio holds promise in remedying the temporal information gap present in vision, and has been explored in several prior works. For example, sound has been shown to estimate the volume of granular material in a container [7], estimate the height of the liquid in pouring [8], identify object properties [9], and be used alongside vision to self-locate [10]. Simultaneously, several works in the computer vision community have shown that audio can even be used to extract visual information

7th Conference on Robot Learning (CoRL 2023), Atlanta, USA.

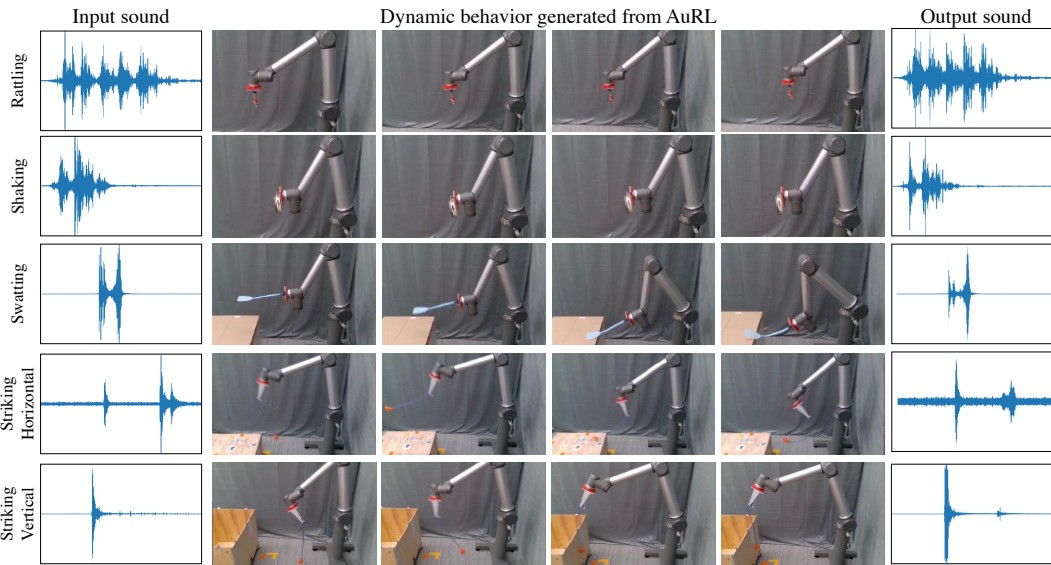

Figure 1: Given desired sounds as input (left column), AURL can generate dynamic, contact-rich behaviors on a UR10 robot (middle) that outputs sound similar to the input (right column). For tasks using multiple microphones, we visualize the waveform only from a single microphone. Note that for several of these dynamic tasks, the visual changes over time are difficult to perceive, while the sound generated by the movement gives a concise signal.

in a scene [11, 12]. The foundation these prior works have built begs the question, can audio also be leveraged to learn contact-rich dynamic manipulation?

In this work, we present AURL, a new learning-based framework for auditory perception in robots that generate sound while interacting with their environment. AURL takes multi-channel sound data as input and outputs parameters that control dynamic behavior. We obtain our sound data from contact microphones placed on and around the robot. Directly reasoning over raw audio data would require gathering large quantities of training data, which limits the use of high-capacity deep networks. To address this, we use self-supervised learning methods to learn low-dimensional representations from audio. These representations form the backbone of AURL and enable efficient learning of dynamic skills with our datasets.

We experimentally evaluate AURL on five contact-rich manipulation tasks including swatting a fly swatter, striking a box, and rattling a child's toy on a UR10 robot (See Fig. 1). Across all tasks, AURL can predict parameters for dynamic behavior from audio with lower errors than our baselines. Furthermore, when given desired sounds to emulate, the dynamic behavior produced by AURL can generate sounds with high similarity to the desired ones. On our sound similarity metrics, we find that self-supervised learning improves performance by 34.5% over supervised learning.

To summarize, this paper makes three contributions. First, we collect an audio-behavior dataset of 25k examples on five tasks that capture contact-rich, dynamic interactions. This dataset will be publicly accessible on our project website. Second, we show that self-supervised learning techniques can significantly improve the performance of behavior prediction from audio. These contact-rich dynamic behaviors are learned solely from audio inputs. Third, we perform an analysis on various design choices such as the self-supervised objective used and the amount of data needed for training.

## 2   Related Work

Our framework builds on several important works in audio-based methods in robotics, self-supervised learning, and dynamic manipulation. In this section, we will briefly describe works that are most relevant to ours.

**Audio for Robotics**: The use of audio information has been extensively explored in the context of multimodal learning with visual data [11, 13, 14, 10, 15]. Several of our design decisions like the use of convolutional networks are inspired by this line of work [11]. In the context of robotics, several works have used audio for better navigation [16, 17, 12, 18], where the central idea is that sound provides a useful signal of the environment around the robot. Recently, the sound generated by a quadrotor has been shown to provide signal for visual localization [19]. The use of audio for manipulation remains sparse [7, 9, 20, 8]. Prior work in this domain has looked at using sound to improve manipulation with granular material [7], connecting sound with robotic planar manipulation [9], and learning imitation-based policies for multimodal sensory inputs [20]. We draw several points of inspiration from these works including the use of contact microphones to record sound [7, 9] and training behavior models [20]. AURL differs from these works in two aspects. First, we focus on dynamic manipulation that generates sound through contact-rich interaction. Second, we show that such manipulation behaviors can be learned without any visual or multimodal data.

**Self-Supervised Learning on Sensory Data**: Self-supervised representation learning has led to impressive results in computer vision [21, 22, 23, 24] and natural language processing [25, 26]. The goal of these methods is to extract low-dimensional representations that can improve downstream learning tasks without the need for labeled data. This is done by first training a model on 'pretext' tasks with an unlabeled dataset (e.g. Internet images). These pretext tasks often include instance invariance to augmentations such as color jitters or rotations [27, 28, 21, 22, 29]. The use of such self-supervised learning has recently shown promise in visual robotics tasks such as door opening [30] and dexterous manipulation [31].

In the context of audio, several recent works have explored self-supervised representation learning for speech and music data [32, 33, 34, 35, 36, 37, 38, 39]. AURL takes inspiration from BYOL-A [32] to use the BYOL [23] framework for learning auditory representations. However, unlike prior works that use music or speech data to train their representations, our data includes multi-channel contact sound that requires special consideration. For example, we found that the Mixup augmentation used in BYOL-A does not work well on our contact sound data.

**Dynamic manipulation**: Training dynamic manipulation behaviors, i.e. behaviors that require dynamic properties such as momentum, has received significant interest in the robotics community [40, 41, 42, 43] . Several works in this domain learn policies that output parameters of predefined motion primitives [44, 45], which makes them amenable to supervised learning approaches. Our work uses a similar action parameterization for the dynamic tasks we consider. However, in contrast to many of these works [46, 45, 47], AURL can operate in domains where visual information does not contain sufficient temporal resolution or contact information to solve desired tasks.

## 3   Audio-Behavior Dataset Collection

Since there are no prior works that have released datasets to study the interaction of audio with dynamic behavior learning, we will need to create our own datasets. We select tasks that allow for self-supervised dataset collection while being dynamic and contact-rich  (see Fig. 1 for task visualization). For tasks that involve making contact with the environment, we use deformable tools to ensure safety. All of our data is collected on a UR10 robot, with behaviors generated through joint velocity control, and sound generated by dynamic contacts recorded by contact microphones.

### 3.1   Dynamic Tasks and Setup

Our dataset consists of five tasks: rattling a rattle, shaking a tambourine, swatting a fly swatter, striking a horizontal surface, and striking a vertical surface. For each task, we attach a different object to our UR10's end effector with 3D printed mounts. Dynamic motion for each task is generated by using motion primitives that control the robot's velocity and acceleration profile. After the execution of the robot motion, distinctive sounds are produced either from the object itself (e.g. rattle) or through interaction with the environment (e.g. swatter). Audio generated from each interaction

is recorded for 4 seconds at 44.1kHz by contact microphones either attached to the robot or in the environment. To visualize behaviors and train vision-based models, we also record RGB visual data from a third-person camera.

For each task, we collect data for $1,000$ unique behaviors, with $5$ examples of each behavior. This collection of multiple examples is done to account for the noise present in audio data even with the same behavior applied. This amounts to $5,000$ datapoints per task; for a total of $25,000$ datapoints in our dataset. Collecting each data point takes ˜10s (including time to safely reset the robot), for a total of ˜14 hours of data collection being run continuously on a robot for each task. To automate the collection of a diverse dataset, first we parameterize the robot's action space then we sample these parameters from a uniform distribution. The sampled primitives are used to generate a trajectory that is then executed on the robot operating in joint velocity control mode. The sampled parameterization of primitives and the self-resetting nature of the tasks allows data collection to proceed with minimal human supervision. For tasks that have parameters that control spatial positions of object interaction, we attach multiple microphones to better capture the spatial relation between sound and action. Details of each of our tasks are as follows:

**Rattling a rattle**: We attach a rattle to the robot's end-effector. The rattle can generate different behaviors through parameters that control the velocities, accelerations, and number of oscillations, to generate rich variations in sound. This task uses a single contact microphone on the rattle to record audio. The primitive is defined by the elbow joint velocity, acceleration and number of oscillations.

**Shaking a tambourine**: Similar to the rattling task, we attach a tambourine to the end-effector. The tambourine can be moved with different velocities, accelerations, and number of oscillations. This task uses a single contact microphone placed on the tambourine attachment. Similar to the rattle, the primitive is defined by the elbow joint velocity, acceleration and number of oscillations

**Swatting a fly swatter**: We attach a flexible fly swatter to the end-effector. The robot then executes a swatting motion on a board placed in front of the robot on the table's top. This task uses two contact microphones on opposite ends of the board to record the audio. The primitive is defined by the velocity of the base joint, the shoulder joint and the acceleration. The acceleration and velocity of the base determines the location of where the flyswatter strikes the board and that of the shoulder joint determines the velocity with which the robot strikes the board.

**Striking Horizontally**: We attach a soft ball to the end of a rope to the robot end-effector. The robot then executes a swinging/dragging motion to strike a plywood board placed horizontally in front of the robot on a closed box. We record the audio produced by this motion with two contact microphones placed on adjacent sides of the box. The action space is defined by the shoulder, elbow and wrist velocities, acceleration, number of time steps for which each joint moves. Varying the number of time steps for which each joint moves allows for more diverse behaviors such as dragging and striking to be generated.

**Striking Vertically**: We use a configuration similar to the striking horizontally task with a deformable ball attached to the end-effector. Instead of a horizontal plane, the robot generates a swinging motion that allows the ball to strike a vertical plywood board. We record the audio produced by this motion with two contact microphones placed on adjacent sides of the vertical surface. The action space is defined by shoulder, elbow and wrist velocities and acceleration. We first apply an initial velocity to generate momentum for the rope. Then we apply a different set of velocities to generate the swinging motion which produces the audio on impact with the open box. Adding momentum allows us to generate more diverse dynamic behaviors.

# 4 AuRL

## 4.1 Overview

Each task in our dataset can be represented as $\mathcal{D} \equiv \{o_k, a_k\}_1^n$, where $o_k$ and $a_k$ represent the audio information and the action information respectively for datapoint $k$. Given this training data, we

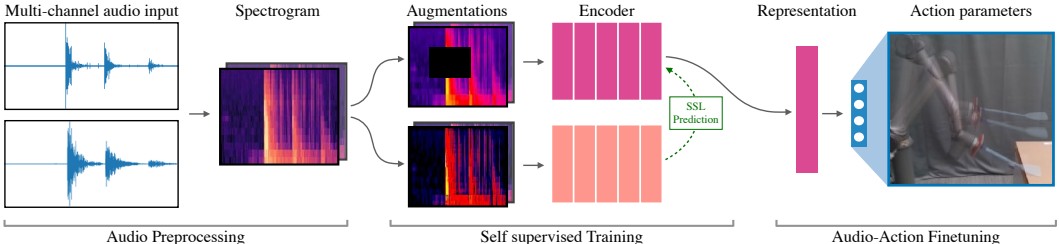

| Multi-channel audio input | Spectrogram | Augmentations | Encoder | Representation | Action parameters |

| Audio Preprocessing | Self supervised Training | Audio-Action Finetuning |

Figure 2: AURL framework: Before training, we preprocess the audio data for compatibility with CNNs. Then, we train an encoder using self-supervised training with BYOL. Finally, we finetune a linear model to predict actions that generate dynamic behaviors on the robot

wish to learn a policy $f(o_k) = \hat{a}_k$ that can predict actions from input sound data. One straightforward approach to learn this is by using standard supervised learning. However, such a learning methodology often requires extensive real-world data that is impractical to collect for many of the dynamic tasks we consider.

We present a two-step approach to efficiently learn with a moderately sized dataset. For each task, we first use state-of-the-art self-supervised learning to train concise aural representations using only our observation data, i.e. $\{o_k\}_1^n$. This model can be represented as $z_k = g_\phi(o_k)$, where $z_k$ is the low-dimensional representation obtained from the high dimensional audio observation $o_k$. Second, we train a supervised linear probe on top of the representation $z_k$, i.e. $\hat{a}_k = w_\theta^\mathsf{T}(z_k)$. We are hence able to leverage high-capacity neural networks in learning representations while preventing overfitting during supervised training by using low-capacity linear models. Our learning framework is illustrated in Fig. 2, with details of individual components in the following sections.

## 4.2 Self-Supervised Representation Learning

Conventionally, learning with audio has been primarily used in the speech and music domains [48, 49]. While several pretrained models leverage audio for speech recognition and music manipulation, there is little work connecting audio to dynamic robotic actions that generate sound. Since the audio from actions differs significantly from speech and music in terms of rhythm, tone, and amplitude, we find that using pretrained audio models does not generalize well to robotic tasks (see Section 5). Hence, to leverage the relatively modest size of training data compared to internet scale audio [50, 51], we rely on self-supervised representation learning.

In this work, inspired by prior work in representation learning for robotics [30, 31], we use Bootstrap Your Own Latent (BYOL) [23] as the self-supervised method to train an encoder. The key idea in BYOL is to learn representations that are robust to augmentations in the input. BYOL does this using two networks, the online network and the target network, which are trained in parallel. The network's objective is to minimize the distance between the representations of different augmented versions of the same audio input. The target network is updated as a slow exponential moving average of the online network to ensure stationarity of the target representation. In our implementation, we normalize the data and use random resize and crop augmentations on the mel spectrogram of the audio inputs.

## 4.3 Audio to Behavior Finetuning

After training the encoder with the self-supervised objective, we use a linear model $w_\theta$ to transform the low-dimensional encodings $z_k$ to the behavior parameters $a_k$. This procedure mimics common practice in computer vision to probe self-supervised models [23, 22, 21]. The linear weights are optimized by gradient descent to minimize the squared loss $\sum_{k=1}^n \|w_\theta^\mathsf{T} z_k - a_k\|^2$. We provide additional training details in Appendix A.

# 5 Experiments

We evaluate AURL on a variety of dynamic manipulation experiments. These experiments are designed to answer the following questions:

- Can AURL learn dynamic behaviors from audio?
- Do AURL behaviors generate desired sounds?
- How important is self-supervised learning in AURL?
- What components and design choices in AURL are essential to performance?

## 5.1 Baseline methods

To contextualize the results from AURL, we run the following methods on our data:

**Random**: Here we select a random action from the training set. This should serve as a lower bound on performance.

**Oracle**: Here each test point is matched to the closest train point, i.e. the train action that has the smallest L2 distance to the test action. This should serve as a rough upper bound on performance.

**Supervised**: Here we randomly initialize a ResNet [52] encoder along with a linear layer as the regression head, and train it from scratch using the MSE objective.

**Supervised+Aug**: Here we follow the same training procedure as the supervised model, but we augment the input data with the same augmentations used in AURL

**Pretrained audio**: Here we extract auditory representations from a pretrained data2vec [53] model, followed by training a linear model to predict the action. This serves as an alternative to the self-supervised pretraining used in AURL.

**Pretrained vision**: Here we extract 5 equally spaced frames from the recorded video of the robot data. The frames are then passed through a frozenImageNet [50] pretrained ResNet [52] encoder to extract representations. We concatenate these representations and learn to predict the action using a linear probe. We only use 5 frames since using a larger number of frames requires significantly larger computational resources compared to the audio models trained in AURL.

## 5.2 Evaluation metrics

To quantify each method, we use the following metrics:

**MSE action prediction**: To quantify the performance of different models we will primarily use the mean squared error (MSE) between the predicted behavior parameters and the ground truth parameters on the test set.

**DTW sound similarity**: To evaluate the similarity between two audio signals, we first take the amplitude envelope of the raw audio signals to remove high-frequency noise. We then use Dynamic Time Warping (DTW) [54, 55] to align the two amplitude envelopes. The distance between the two envelopes after DTW will serve as our raw distance metric. We then normalize this score, as $(x - \mu)/\sigma$, where $x$ is the distance, $\mu$, and $\sigma$ is the mean and standard deviation of the raw distance values of sounds generated by the same action applied on the robot.

We note that this representation for similarity deviates from metrics in music and speech recognition such as chromagrams[56] or MFCCs[57]. This is because metrics in music focus on the temporal nuances (pitch and tone) in sharp sounds created by our robots and tend to ignore the general shape of the audio waveform. In contrast, the DTW metric does not assume that the signals have the same length or are even aligned in time. Instead, it aligns the two signals in time by warping one of the signals so that it best matches the other signal. This makes it suitable for tasks where the signals may have different lengths or may be shifted in time, both of which are expected when comparing two audio signals.

Table 1: MSE error (↓) on the prediction of behavior parameters for baselines and AURL

| | Rattling | Shaking | Swatting | Striking (H) | Striking (V) |
|---|---|---|---|---|---|
| **Random** | 2.36 | 2.61 | 0.67 | 1.10 | 1.06 |
| **Oracle** | 0.07 | 0.07 | 0.06 | 0.31 | 0.32 |
| **Sup** | 0.25 | 0.28 | 0.09 | 0.59 | 0.45 |
| **Sup+Aug** | 2.37 | 2.19 | 0.27 | 0.76 | 0.58 |
| **Pre. Aud** | 3.10 | 4.19 | 0.25 | 0.65 | 0.53 |
| **Pre. Vid** | 2.30 | 3.05 | 0.05 | **0.52** | 0.48 |
| **AURL** | **0.12** | **0.22** | 0.02 | **0.53** | 0.39 |
| **AURL+Act** | 0.12 | 0.23 | 0.02 | 0.58 | 0.41 |
| **AURL+AA** | 0.14 | 0.26 | **0.01** | 0.54 | **0.38** |
| **AURL+MT** | **0.12** | **0.15** | **0.01** | 0.60 | 0.43 |

### 5.3 Can AURL learn dynamic, contact-rich behaviors from audio?

To quantitatively evaluate the behaviors learned, we compare the MSE between the action predicted by the learned model and the ground truth action used to generate the behavior on the robot. A summary of these results with the MSE metric is in Table 1. We see that the self-supervised methods outperform other competitive approaches. Particularly for the Striking Vertically and Swatting tasks, we find that our trained models can achieve a performance very close to the Oracle method. Interestingly, we find that supervised learning with augmented data performs quite poorly, which highlights the importance of using self-supervision to extract information from augmentations.

### 5.4 Do the AURL behaviors generate desired sounds?

To gauge the dynamism of the produced behaviors and to ensure fidelity to the input audio, we run our trained models on our UR10 robot and measure the similarity of the generated sounds with the input desired sound (see Appendix B) using the normalized DTW metric introduced in 5.2. We summarize the results of this experiment in Table 2. We observe that AURL methods outperform baselines on 4/5 tasks. In the Shaking task, supervised training performs slightly better than the self supervised methods. In tasks such as Striking(H) and Rattling, we notice that AURL can outperform the Oracle method, which indicates that self supervision can be useful in extracting good representations. This demonstrates the promise of self-supervision to interpolate and select actions that are not present in the training set.

### 5.5 How important is self-supervised learning in AURL?

Instead of using the self-supervised objective in AURL, one could also use pretrained representational models that were trained on much larger, albeit out-of-distribution data [50, 58]. We compare against two such pretrained models. The first uses pretrained audio models from data2vec [58], and the second uses pretrained visual models learned from ImageNet data. As we see in Table 1 both these pretrained models perform quite poorly. This is not surprising since the data used to pretrain the audio models consists of music and speech data, while our task requires understanding contact sounds. Similarly, image-based models due to their low temporal frequency cannot capture information relevant to predicting dynamic behavior.

### 5.6 What components and design choices in AURL are essential to performance?

We experiment with several design choices from our dataset collection setup to our learning framework.

**Contact microphones**: At the hardware level we found that using contact microphones instead of standard microphones allowed us to record sounds with significantly lesser ambient noise (e.g. people speaking in the background). This reduces the need for post-processing to denoise the data.

Table 2: Normalized DTW distance (↓) between the amplitude envelopes of the desired audio and produced audio.

| | Rattling | Shaking | Swatting | Striking (H) | Striking (V) |
|---|---|---|---|---|---|
| **Random** | 4.95 | 8.33 | 7.12 | 6.05 | 3.36 |
| **Oracle** | 1.07 | 7.61 | 4.47 | 3.24 | 1.85 |
| **Sup** | 1.10 | **7.67** | 5.86 | 2.43 | 2.53 |
| **Sup+Aug** | 6.51 | 8.22 | 5.80 | 3.81 | 2.37 |
| **AᴜRL** | **0.96** | 7.76 | 5.23 | 2.35 | **2.05** |
| **AᴜRL+AA** | 0.97 | **7.68** | **4.98** | **2.18** | 2.13 |

**Scaling with data size**: We studied the effects of data scale with performance by running our method across a variety of different slices of training data collected in AᴜRL. We find that although increasing the amount of data helps, self-supervised pretraining leads to larger relative improvements in the low data regime. (See Appendix C for more details). Hence, like several large-scale machine learning works [26, 52], it leads us to conjecture that while at the limit of data size plain supervision may be sufficient, it is in the low and moderately sized data regimes in which specialized self-supervision techniques might be crucial. Further, collecting a large dataset is time intensive. For better performance with limited data, self supervised learning may be beneficial.

**Types of augmentation**: We found that the standard audio augmentation of Mixup [32] performs quite poorly with our data. This is understandable since mixing different audio signals loses the temporal information needed to predict behaviors. Similar to recent works in reinforcement learning [59] we find that using just a few augmentations (random resize and crop) works quite well for contact sound data. We note that commonly used image augmentation techniques like rotation and gaussian blur are not meaningful when applied to audio data (spectrograms).

**Mechanisms of self-supervision**: Instead of running standard BYOL, our control over the dataset generation process allows us to collect data that explicitly contains the invariances we require from auditory representations. For example, instead of specifying the invariance through data augmentations, we can specify the invariance by using two audio signals generated by the same behavior. As denoted by AᴜRL+Act in Table 1 we see that although we obtain gains against supervised methods it does not perform as well as AᴜRL. However, we find that using this invariance along with augmentations, denoted as AᴜRL+AA, improves aural similarity performance on 3/5 tasks.

**Multi-task pretraining**: To evaluate if task specific pretraining is beneficial for AᴜRL, we pretrain the AᴜRL encoder using standard BYOL on all of the audio data collected across all tasks. As denoted by AᴜRL+MT in Table 1 we see that AᴜRL+MT performs at par with other AᴜRL methods for 2/5 tasks and performs better in only 1/5 tasks. One reason for this is that the striking tasks show sparse sharp spikes in the waveform/ spectrogram while rattling, shaking and swatting show dense spikes in the spectrogram. Due to the large variation in the distributions of the spectrograms between tasks we posit that task-specific pretraining is ideal for our set of tasks.

## 6   Limitations and Discussion

In this paper we have presented AᴜRL, a framework to train self-supervised auditory representations for dynamic manipulation with contact. While we show improved results with audio inputs, there are currently two limitations of this work. First, for tasks such as Striking Horizontally and Shaking a Tambourine we find only mild gains over plain supervised learning on the MSE action prediction metric, while being significantly poorer than the Oracle method. Further research in improving robotic self-supervision or leveraging large-scale audio data can significantly improve these models. Second, our current framework focuses primarily on learning the interactions from labeled datasets, which limits our models to only capture behaviors present in our training set. Expanding AᴜRL to reinforcement learning based dataset collection can allow for task-specific learning.

## Acknowledgments

If a paper is accepted, the final camera-ready version will (and probably should) include acknowledgments. All acknowledgments go at the end of the paper, including thanks to reviewers who gave useful comments, to colleagues who contributed to the ideas, and to funding agencies and corporate sponsors that provided financial support.

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

# A Training Details

**Audio preprocessing**: Since we use convolutional networks to fit our audio-action data, we will need to first process the audio to have compatible dimensions. To ensure that all audio data have the same length of 4s, we clip longer sequences and pad shorter audio sequences to equal lengths. Next, we downsample the audio from 44.1kHz to 11kHz to reduce the dimensionality of the data. To make our audio data compatible with CNNs, we can either use a plain STFT [60], mel, or log-mel [61] spectrogram. We choose the mel spectrogram representation over the other options based on initial experiments. To do this, a STFT of window size 400 and hop length 160 is first applied on the downsampled audio, followed by the application of a mel filter [62] with 16 mel banks. To ensure compatibility with a different number of microphones, we treat data from each microphone as a separate channel. This representation allows us to treat 1D audio observations as 2D images.

**Self-supervised training**: For all of our experiments with self-supervised learning, we use a ResNet-18 [52] encoder with the number of channels equal to the number of contact microphones used for the corresponding task. The length of the latent representation $z_k$ for all tasks is fixed to 512. Following standard practice [52], we normalize the mel-spectrograms of the audio with channel-wise mean and variance. We train each encoder for 1000 epochs on the task's audio data. We use the LARS [63] optimizer with an initial learning rate of 0.2, weight decay of $1.5e^{-6}$, and a batch size of 1024.

**Finetuning**: For each of our tasks, we split our dataset of 5000 datapoints into 4000 train audio-action pairs and 1000 unseen test audio-action pairs. We train the linear weights $w_\theta$ with a batch size of 1024, the Adam [64] optimizer with a learning rate of $10^{-4}$, and a weight decay of $10^{-4}$.

We normalize our inputs with mean and standard deviation of the each channel. For the output of each of our models, for all tasks except shaking and rattling, to predict the actions, we perform min max normalization of actions from the train dataset. While evaluating our model on the robot, we scale the predicted actions with the minimum and maximum calculated during training. For tasks that do not use a normalized action space, we clip the maximum and minimum of the action space. We do so to ensure that the robot operates within safety limits.

# B Qualitative Results

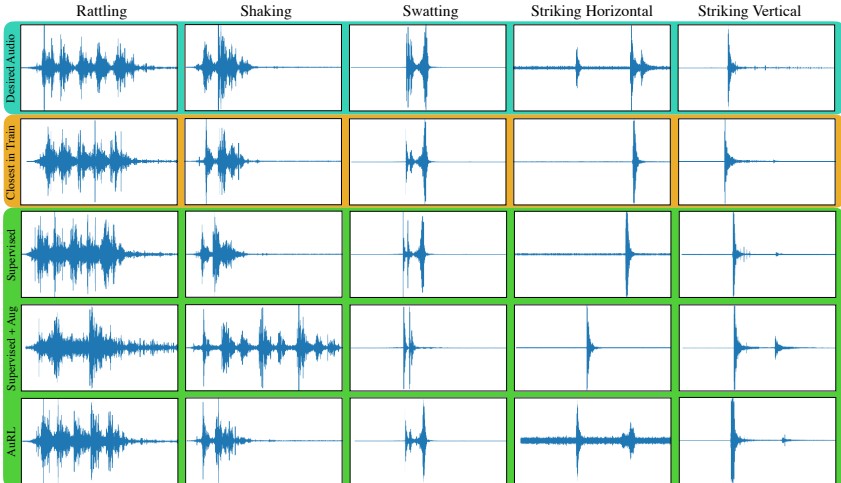

Figure 3: Waveforms of audio generated by our robot during the evaluation of different models. For tasks using multiple microphones, we visualize the waveforms from a single microphone. We see that the models learned with AURL better match desired audio compared to standard supervised approaches, particularly on the Striking Horizontal task.

## C   Scaling effects of dataset size

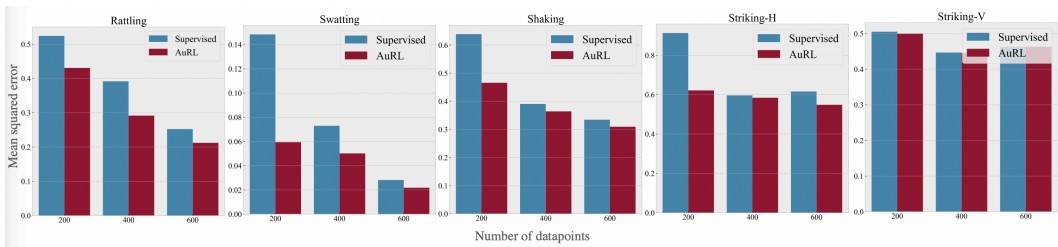

Figure 4: Comparison of AURL and supervised learning on MSE error (↓) when varying the number of unique actions used in training. We see that in the low data regime, self-supervised pretraining through AURL significantly improves upon vanilla supervised learning. However, there are diminishing returns when the dataset size increases.

