# OpenReview forum: "That Sounds Right: Auditory Self-Supervision for Dynamic Robot Manipulation"
_robot-learning.org/CoRL/2023/Conference — CoRL 2023 Poster_

### Official Review · Reviewer_KfWt · 2023-07-18

**Confidence:** 4
**Originality:** Good
**Technical Quality:** Good
**Clarity Of Presentation:** Good
**Impact:** 3

**Recommendation:**

Weak Accept: I recommend accepting the paper, but will not argue for my recommendation if the majority of other reviewers have a different opinion.

**Review:**

Strengths:
*The authors presents a novel audio dataset for robotic manipulation.
*Sufficient experiments are conducted on five different tasks to demonstrate the increase in performance.

Weaknesses:
*Limitation in the use of this dataset. It would be more interesting to see if pretraining on this dataset will have performance increase in some more complex robotic manipulation tasks(eg. peg in hole task, training from scratch vs. pretraining on this dataset).

**Quality Of The Limitations Section:**

Additional details required

**Questions For Rebuttal:**

The audio dataset for contact-rich manipulation is very interesting and novel. My only thought is that it would be more interesting to see if this dataset can be used to improve the performance of other more complex manipulation tasks.

There are previous work that leverage audio to learn contact-rich dynamic manipulation (See, Hear, and Feel by Li and Play it by ear by Du). But their limitations are that they are collecting their own task-specific audio dataset and train their model from scratch. I think some of the behaviors in your paper are representative in contact-rich manipulation tasks (like the collision sound, it would be better to have some scratch sound as well). It might have bigger impact to the community if this dataset can be generalized to other tasks and other people can used it for pretraining.

**Robotics Focus:**

Sufficient demonstration on hardware

**Summary Of Paper:**

In this paper, the authors collect a audio dataset and use self-supervised learning to improve performance of behavior prediction from audio.

**Summary Of Recommendation:**

I think the framework is interesting and the dataset is quite novel since it is a visual audio dataset for robotic manipulation. Hence, I recommend accept the paper.

---

### Official Review · Reviewer_KaUb · 2023-07-18

**Confidence:** 3
**Originality:** Good
**Technical Quality:** Fair
**Clarity Of Presentation:** Fair
**Impact:** 3

**Recommendation:**

Weak Accept: I recommend accepting the paper, but will not argue for my recommendation if the majority of other reviewers have a different opinion.

**Review:**

Globally this work lacks many details and rigorousness, which confuse the reader and drastically reduce the quality of the paper.
Strengths:
- Original and relevant research area.
- Experimental validation of the approach

Weaknesses:
- Unclear methodology. In baseline supervised/unsupervised, It is crucial for the reader to understand from the start  the inputs and outputs of the proposed method. Yet, after reading the paper, the reader is unable to understand the output of the proposed method.  How is the robot controlled (discrete primitives? velocity control? sequence?).  This is a major flaw.
- Many inaccurate/undefined terms in equations
- Experimental evaluation unclear: The MSE is computed for which behavior parameters?  What is the unit? How is the L2 distance computed? What is a test point?
- The organization of the experimental results is confusing. It start from Table 1 to Table 2 then back to Table 1...

**Quality Of The Limitations Section:**

Additional details required

**Questions For Rebuttal:**

1) Please specify clearly the inputs and outputs of the proposed method.
2) Please detail the method figure. The figure is too shallow.
3) Line 161, $a_k$ is not a policy from the reviewer understanding. It's the output. The policy is $f$.
4) Line 167, what are the purposes of the indexes $1$ and $n$?  The same were used in line 159 but never defined.
5) Line 169, what is the meaning of $\theta$
6) Line 194, rigorously the squared loss function is wrong.  With the author's notation this would be equal to 0, because of the definition of $a_k$ line 169. Again, somewhere should be defined the ground truth and predicted value. This notation error principally comes from the fact that the input and outputs were not correctly and clearly specified.
7) Please improve the clarity  and organization of the experimental section  as mentioned in the weaknesses above.
8)The authors overlooked/did not mention clearly,  that the BYOL encoder is trained separately for each different task. Each task seems to have different number actions (i.e. output size). So a custom AuRL network is needed for each task. This a  clear practical limitation in the reviewer opinion.
9) Please detail the intra-task data distribution ? What are the different sound collected from the same task ?

**Robotics Focus:**

Sufficient demonstration on hardware

**Summary Of Paper:**

This paper introduce AuRL, a self-supervised network for generating robot motion from a sound input.
AuRL is composed of two parts: a self-supervision/encoder network that defines latent audio features; a fine tuning network that predicts the motion to generates a similar sound as the input.

**Summary Of Recommendation:**

Given the many technical and experimental flaws, this paper is not currently at the level of a CoRL paper.
It is highly recommended to check the soundness of all equations, improve the clarity of the paper and provide additional details about the experimental study.

Post Rebuttal Update:
The reviewer appreciates the efforts and rigorous answers of the authors in the rebuttal process.  The reviewer regrets that the same rigorousness and efforts were not put in the first version of the paper (with obvious "amateurish" flaws). If the paper is accepted, it requires some substantial reworking from version 1. Despite the potential benefit for the robot audition community, because of the above  weaknesses, the reviewer updates his recommendation to weak accept.

---

### Official Review · Reviewer_UsSe · 2023-07-19

**Confidence:** 3
**Originality:** Good
**Technical Quality:** Good
**Clarity Of Presentation:** Fair
**Impact:** 3

**Recommendation:**

Weak Accept: I recommend accepting the paper, but will not argue for my recommendation if the majority of other reviewers have a different opinion.

**Review:**

Strengths:
- The paper is well-written and easy to comprehend
- The task of learning context-rich tasks through auditory inputs is interesting.

Weaknesses:
- The ablation study provided within the paper lacks compelling evidence
- I'm not fully convinced with the setting of using audio inputs without combining them with visual inputs. The motivation behind this approach needs to be clarified.

**Quality Of The Limitations Section:**

Limitations are addressed clearly

**Questions For Rebuttal:**

- The 'Oracle' method, as it is presented in the paper, appears to be a nearest-neighbor approach. As per my understanding, it operates by identifying the closest audio sample within the training data given a desired auditory target input, and then mimics the corresponding action from the training sample. If this is the case, the use of the term 'Oracle' seems to be misleading, as this approach doesn't take more inputs than other methods. From the experiment results, it seems that this conventional nearest-neighbor method is better than training neural networks.

- The paper's motivation to use auditory-only inputs is unconvincing. The introduction of multi-modal inputs could potentially enhance task completion. Most manipulation tasks, particularly context-rich ones, greatly benefit from visual inputs, making them essential for learning actions.

- Building upon the previous point, the paper claims that the tasks under investigation cannot be accomplished using solely visual inputs. To support this claim, the author should provide a more thorough evaluation of the visual baselines used for comparison. Several critical technical details remain undisclosed. For instance, what is the duration of the five frame inputs used in the vision baseline, and how does this compare to the audio inputs? Is the ImageNet pretrained ResNet features frozen during the process or are they fine-tuned?

**Robotics Focus:**

Sufficient demonstration on hardware

**Summary Of Paper:**

This paper explores the concept of motor learning through the use of auditory inputs. The tasks examined are those typically underperformed when relying solely on visual data. This includes actions like rattling a rattle or shaking a tambourine, where the completion of contact-rich tasks is heavily reliant on sound. The authors collected a robust dataset comprising 5000 data points per task. The tasks evaluated include a range of activities including rattling a rattle, shaking a tambourine, swatting with a fly swatter, and performing horizontal and vertical strikes using a UR10 robot. The proposed model outputs actions given a desired sound as input.

The proposed model transforms the audio signal into a spectrogram, then a CNN is used to extract features. These features are trained using the BYOL target, before a linear model translates these features into corresponding actions.
An important finding in this study was the use of self-supervised learning using the BYOL method.

**Summary Of Recommendation:**

This paper introduces an novel approach to motor learning, where actions are learned from desired audio inputs. This novel setting could potentially contribute fresh insights to the field of multi-modal robot manipulation, which leads me to recommend acceptance. However, I must point out that the experiments presented within the paper do not fully substantiate the claims made by the authors. Additionally, the use of a self-supervised learning approach during feature extraction is not particularly groundbreaking. I would suggest the authors to enhance their experimental design and offer more compelling evidence to support their claims.

-------------
Post rebuttal: My questions has been addressed and I recommand weak accept

---

### Official Review · Reviewer_N4ak · 2023-07-20

**Confidence:** 3
**Originality:** Excellent
**Technical Quality:** Very Good
**Clarity Of Presentation:** Very Good
**Impact:** 4

**Recommendation:**

Weak Accept: I recommend accepting the paper, but will not argue for my recommendation if the majority of other reviewers have a different opinion.

**Review:**

I like this paper, its very humorously written and is a surprisingly innovative approach to robot learning. That is refreshing. I am not an expert in audio tech but the methods used look reasonable to me.
1. For the table 1, the abbreviation for methods in the lower blocks, it is not apparent to me what they stand for. Would be nice to explain below.
2. The result that self supervised learning is better than supervised pretraining in data constrained regimes is an important observation.
3. Limitations and discussions sections could be improved
4. An interesting question is, can we do audio conditioned task execution using similar methods or speech-to-actions?


**Quality Of The Limitations Section:**

Limitations are addressed clearly

**Questions For Rebuttal:**

1. Why is the random baseline better than pretrained audio baseline? That's weird?
2. The website doesn't have qualitative results. MSE is a good metric but it is not a palpable metric, some examples of sounds produced by robot after learning could be useful to understand where things stand.
3. What are the main failure modes of this approach? A little bit on this would be useful.
4. Would the self supervision result generalize to generic data constrained regimes?

**Robotics Focus:**

Sufficient demonstration on hardware

**Summary Of Paper:**

The paper pushes robot learning using audio input. Given a sound, a robot is tasked with creating the sound with behaviours. They use self supervised learning technique to learn latent representations of audio, then a second network to learn behaviour given a latent representation. Mean square error over produced sound is used as an accuracy measure. The paper benchmarks against a few known baselines and does ablations over supervised learning and a couple other design choices.

**Summary Of Recommendation:**

I like this paper, it's an intersecting problem set up, an unusual and infrequently explored data modality. the hypotheses generated are sound and the experiments make a reasonable shot at answering them. Qualitative results on actual audio samples will greatly improve this paper. I would accept this paper, so more roboticists can think about working with audio data.

---

### Author Response · Authors · 2023-08-09
**General response, list of updates**

Thank you for your thoughtful feedback and for finding our work interesting (KfWt) and  innovative (N4ak). In this global comment we will address concerns about our motivation behind using audio for dynamic, contact-rich manipulation and discuss the current limitations of our work. We follow this by a brief description of changes made to the manuscript to address concerns raised by the reviewers.

### Motivation behind using audio
In this section, we address concerns from reviewer UsSe about our motivation to use audio.
- Imagine swatting a flyswatter. The force with which we use the fly swatter is not immediately apparent from visual inputs, but can easily be discerned from the sharp thwack noise.
- According to the suggestion made by reviewer UsSe, multimodal training would provide better representations. In our work, we choose to experiment with using a smaller model to extract useful information while using a simple but underused sensory modality. We view the use of small, simple but effective models as a strength in our paper. Using video for the set of tasks we provide in our paper would require more computational resources. Further, multimodal training which requires significantly more computational power can often be hard to train and is an open research question in itself.

### Limitations of our approach
In this section we address concerns about the limitations of our work raised by reviewers N4aK and UsSe.
- **Failure modes**: Generalizing learned behaviors to a different spatial orientation would be challenging.
- **Pretrained models**: Unlike work with visual data where pretrained models improve performance of downstream tasks, we find that existing pretrained models for audio focus primarily on rhythmic sounds like speech or music. Creating better pretrained audio models for robot tasks is an open challenge.

We updated the manuscript with the following modifications to incorporate your feedback into our work.

We make the following changes suggested by reviewer KaUb to improve the clarity of the equations.

- In line 161, we update the definition of policy f which takes in inputs of observation (audio) $o_k$ and outputs predicted action $\hat{a}_k$. This change will also correct the squared loss function on line 194

We hope that these clarifications and updates to our paper inspire further confidence in our work. We invite any further questions or feedback that you may have on this paper.

---

### Decision · Program_Chairs · 2023-08-30

**Decision:**

Accept (Poster)

**Comment:**

This submission initially received mixed reviews.  After discussion, the reviewers generally leaned toward acceptance based on the technical innovations and the experiments.  The AC agrees.  The authors are encouraged to further revise the paper based on the reviews in the camera-ready version.